# Approximate Feature Collisions in Neural Nets

**Ke Li**[*]
UC Berkeley
ke.li@eecs.berkeley.edu

**Tianhao Zhang**[*]
Nanjing University
bryanzhang@smail.nju.edu.cn

**Jitendra Malik**
UC Berkeley
malik@eecs.berkeley.edu

## Abstract

Work on adversarial examples has shown that neural nets are surprisingly sensitive to adversarially chosen changes of small magnitude. In this paper, we show the opposite: neural nets could be surprisingly *insensitive* to adversarially chosen changes of large magnitude. We observe that this phenomenon can arise from the intrinsic properties of the ReLU activation function. As a result, two very different examples could share the same feature activation and therefore the same classification decision. We refer to this phenomenon as feature collision and the corresponding examples as colliding examples. We find that colliding examples are quite abundant: we empirically demonstrate the existence of polytopes of approximately colliding examples in the neighbourhood of practically any example.

## 1 Introduction

Deep learning has achieved resounding success in recent years and is quickly becoming an integral component of many real-world systems. As a result of its success, increasing attention has focused on studying the robustness of neural nets, in particular to inputs that are deliberately chosen to yield an unexpected prediction.

It is well-known that neural nets could be surprisingly *sensitive* to *minute* changes to the input that are deliberately designed to result in a different classification decision. Such examples are known as *adversarial examples* and have been extensively studied in the literature (Dalvi et al., 2004; Biggio et al., 2013; Szegedy et al., 2013). In this paper, we demonstrate the existence of a phenomenon that is in some sense the *opposite*: we show that neural nets could be surprisingly *insensitive* to *large* changes to the input. We provide an explanation for how and why this could arise and propose a method for systematically finding such kinds of inputs.

To find changes that a neural net is *insensitive* to, it suffices to find examples that share the same feature activation at some layer, since the same activation at some layer implies the same activation at all subsequent layers and therefore the same classification decision. We will refer to the phenomenon of multiple examples sharing the same feature activations as feature collisions [2] and such examples

---

[*]Equal contribution.

[2]It should noted that the term "feature collision" was also used by (Shafahi et al., 2018) to refer to a different, but related, concept. In their context, feature collisions refer to input examples that are close to *both* a target data example in feature space and a different data example from a different class in input space. In our context, input examples only need to be close to a target data example in feature space; they do not need to be close to any particular data example in input space.

as colliding examples, to borrow terminology from the hashing literature, since feature collisions correspond to cases where the neural net maps different inputs to the same activation.

A neural net is essentially "blind" to the differences between colliding examples, and so one would hope that such examples are rare and isolated, and when they do occur, are very similar to each other so that differences between them can be safely ignored. We show in this paper that for common neural net architectures, such examples could actually be quite abundant – in fact, there could be a polytope containing infinitely many colliding examples, that is, *all* examples inside this polytope have the same feature activation. We show this arises from the geometry of rectified linear unit (ReLU) activation functions. The key observation is that we could change any negative component in the pre-activation of a layer before ReLU to any other negative value without changing the post-activation, and consequently keep the pre- and post-activations in all later layers the same. We show that in general, for any neural net with a layer with ReLU activations, there is a convex (but possibly unbounded) polytope in the space of post-activations of the previous layer such that all input examples whose post-activation vectors fall in the interior or the boundary of the polytope will have identical activations in all later layers. We devise a method of empirically finding relaxed versions of such polytopes, within which feature activations are approximately equal.

It turns out such polytopes are surprisingly common. We demonstrate the existence of such polytopes in the neighbourhood of practically *any* example that we tried. Furthermore, we demonstrate the existence of such polytopes regardless of what target feature activation the examples inside the polytope are made to collide with. We find that the radius of the polytopes is quite large. Moreover, we empirically confirm that *all* of 2000 examples randomly drawn from the polytope are classified confidently into the same class.

Due to humans' insensitivity to differences at high frequencies, large differences in terms of magnitude may not be perceptually obvious. To find polytopes where images at different locations are perceptually different, we also constrain the search space to compositions of image patches and demonstrate the existence of such polytopes even under this constrained setting.

Because such polytopes arise from the properties of the architecture itself (in particular the activation functions) rather than the weights, the polytopes cannot be eliminated by simply augmenting the dataset. Training the neural net on a different dataset can only make the weights of the neural net different and so cannot in general change the *existence* of the polytopes (though it may change the volume of the polytopes).

## 2    Method

Consider any layer in a neural net with ReLU activations. Let $\mathbf{W} \in \mathbb{R}^{N \times d}$ denote the weight matrix, $\mathbf{x} \in \mathbb{R}^d$ the previous layer's post-activation, $\mathbf{b} \in \mathbb{R}^N$ the biases associated with the current layer. Moreover, let's define $\mathbf{y} \in \mathbb{R}^N$ as the vector of post-activations (activations after the ReLU), and $\tilde{\mathbf{y}} \in \mathbb{R}^N$ as the vector of pre-activations (activations before the ReLU). So,

$$\mathbf{y} = \max(\tilde{\mathbf{y}}, 0) = \max(\mathbf{W}\mathbf{x} + \mathbf{b}, 0), \text{ where } \mathbf{W} = \begin{bmatrix} \mathbf{w}_1^\top \\ \vdots \\ \mathbf{w}_n^\top \end{bmatrix} \quad \mathbf{b} = \begin{bmatrix} b_1 \\ \vdots \\ b_n \end{bmatrix}$$

Our goal is to find a colliding example that has the same post-activations as a target example. We can identify examples by their post-activations in the previous layer, since two examples with identical post-activations in the previous layer will always have identical pre- and post-activations in the current layer. We will denote the colliding example as $\mathbf{x}^*$ and the target example as $\mathbf{x}^t$ and define $\mathbf{y}^*, \tilde{\mathbf{y}}^*, \mathbf{y}^t, \tilde{\mathbf{y}}^t$ analogously. For any vector $\mathbf{v}$, we will use the notation $\mathbf{v}_i$ to denote the $i$th component.

Since ReLUs map all non-positive values to zeros, if the target example has a post-activation of zero in one component, as long as the pre-activation of the colliding example is non-positive in that component, then the target and the colliding example would have the same post-activation in that component. In order to make the post-activations of the two examples identical in all components, the following conditions are necessary and sufficient:

$$\forall i \text{ such that } \tilde{\mathbf{y}}_i^t > 0, \tilde{\mathbf{y}}_i^* = \mathbf{w}_i^\top \mathbf{x}^* + b_i = \tilde{\mathbf{y}}_i^t$$
$$\forall i \text{ such that } \tilde{\mathbf{y}}_i^t \leq 0, \tilde{\mathbf{y}}_i^* = \mathbf{w}_i^\top \mathbf{x}^* + b_i \leq 0$$

**Proposition 1.** *Let $N^+$ be the number of components in $\tilde{\mathbf{y}}^t$ that are positive. Let $\mathbf{W}^+ \in \mathbb{R}^{N^+ \times d}$ and $\mathbf{W}^- \in \mathbb{R}^{(N-N^+) \times d}$ be the submatrices of $\mathbf{W}$ consisting of rows where the corresponding components in $\tilde{\mathbf{y}}^t$ are positive and non-positive respectively. Assume $N^+ < d$. Let $\ker(\cdot)$ denote the kernel of a matrix. The set of colliding examples forms a convex (but possibly unbounded) polytope in at least a $d - N^+$-dimensional (affine) subspace if $\ker(\mathbf{W}^+) \not\subseteq \ker(\mathbf{W}^-)$, or a subspace of at least $d - N^+$ dimensions otherwise.*

*Proof.* The set of colliding examples must satisfy all of the above conditions, so it is the intersection of the following sets:

$$\{\mathbf{x}^* : \mathbf{w}_i^\top \mathbf{x}^* + b_i = \tilde{\mathbf{y}}_i^t\} \text{ for } i \text{ such that } \tilde{\mathbf{y}}_i^t > 0$$
$$\{\mathbf{x}^* : \mathbf{w}_i^\top \mathbf{x}^* + b_i \leq 0\} \text{ for } i \text{ such that } \tilde{\mathbf{y}}_i^t \leq 0$$

Geometrically, each set that corresponds to an equality constraint represents a $(d-1)$-dimensional hyperplane. Each set that corresponds to an inequality constraint represents a half-space. The conjunction of the equality constraints is the intersection of the associated hyperplanes, which is an affine subspace of at least $d - N^+$ dimensions. We consider the projection of each half-space onto this subspace, which is either a half-space in the subspace or the subspace itself.

When $\ker(\mathbf{W}^+) \not\subseteq \ker(\mathbf{W}^-)$, at least the projection of one half-space is a half-space in the subspace (see the appendix for a derivation of this fact). The set of colliding examples is the intersection of all projections of half-spaces, which is the intersection of all projections that are half-spaces in the subspace. In general, the intersection of finitely many half-spaces is a convex (but possibly unbounded) polytope. So, in this case, the set of colliding examples is a convex polytope in the subspace.

When $\ker(\mathbf{W}^+) \subseteq \ker(\mathbf{W}^-)$, all projections of half-spaces are the subspace itself, and so the intersection of the projections is simply the subspace itself. Therefore, in this case, the set of colliding examples is the subspace. $\square$

Any point in this polytope corresponds to a colliding example. Since there could be infinitely many points in a polytope, there could be infinitely colliding examples. If the polytope is bounded, we would be able to characterize all such examples by the vertices of the polytope, in which case the polytope would simply be the convex hull of all the vertices. Then, any colliding example can be written as a convex combination of the vertices, and we can generate a new colliding example by taking an arbitrary convex combination of the vertices.[3]

To demonstrate the existence of this polytope, it suffices to find a subset contained in the polytope, e.g.: the convex hull of a subset of the vertices. To find a subset of vertices, we can move the dividing hyperplane of a half-space towards the feasible direction, which mathematically corresponds to decreasing the RHS of the corresponding inequality constraint. This is equivalent to picking a unit in the current layer and trying to make it as negative as possible. This process is illustrated in Figure [1] in 2D. To find a different vertex, we can simply pick a different constraint to optimize.

To find a vertex, we solve the following optimization problem for some $i$ such that $\tilde{\mathbf{y}}_i^t \leq 0$. Note that instead of searching over the space of the previous layer's post-activations $\mathbf{x}$, we can directly search over the space of inputs $\mathbf{u}$. Let $\tilde{\mathbf{y}}(\mathbf{u})$ and $\mathbf{y}(\mathbf{u})$ denote the current layer's pre-activation and post-activation of an arbitrary input $\mathbf{u}$.

$$\min_{\mathbf{u}} \alpha \mathcal{L}_+(\mathbf{u}) + \beta \mathcal{L}_-^i(\mathbf{u}), \text{ where } \mathcal{L}_+(\mathbf{u}) = \frac{1}{2} \left\| \tilde{\mathbf{y}}(\mathbf{u}) \odot \mathbf{1} \left[ \tilde{\mathbf{y}}^t > 0 \right] - \mathbf{y}^t \right\|_2^2$$
$$\text{and } \mathcal{L}_-^i(\mathbf{u}) = \left( \tilde{\mathbf{y}}(\mathbf{u}) \odot \mathbf{1} \left[ \tilde{\mathbf{y}}^t \leq 0 \right] \right)_i + \sum_{j \neq i} \left| \left( \tilde{\mathbf{y}}(\mathbf{u}) \odot \mathbf{1} \left[ \tilde{\mathbf{y}}^t \leq 0 \right] \right)_j \right|$$

Intuitively, $\mathcal{L}_+$ aims to keep the coordinates in which the target example has positive pre-activations similar between the target and the colliding example. On the other hand, $\mathcal{L}_-^i$ places constraints on the coordinates in which the target has non-positive pre-activations. The first term of $\mathcal{L}_-^i$ tries to make the pre-activation of the selected hidden unit $i$ as negative as possible and the second term keeps the

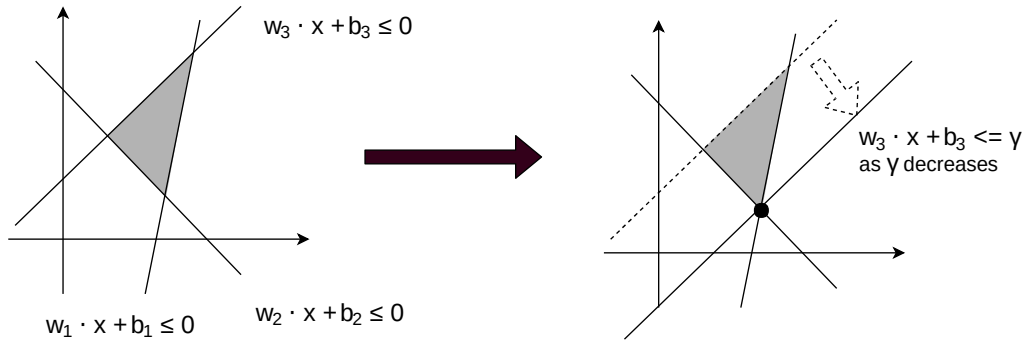

Figure 1: On the left, we illustrate the polytope that arises when there are three half spaces in a two-dimensional subspace. On the right, we illustrate how we can find a vertex of a polytope.

other hidden units close to zero. It does not matter how negative the selected unit becomes, since after applying the ReLU activation function, it becomes zero, which by definition is equal to the post-activation of the target example.

## 3  Experiment

In this section we will apply our method to two standard neural net architectures trained on the MNIST and ImageNet datasets.

### 3.1  MNIST Dataset

First we train a simple fully-connected neural network with two hidden layers [4], each with 256 units and ReLU activations on the MNIST dataset. The trained model achieves a test accuracy of 96.64%.

Next, we apply the proposed method to find feature collisions at the first hidden layer. We performed two experiments and found two polytopes that collide with the same target example, one by initializing from the target example, and one by initializing from a different example in the dataset. For each experiment, we found five vertices of the polytope, by optimizing the proposed objective for $i = 0, 1, \ldots, 4$. The results are shown in Table 1.

Any example within the convex hull of the five vertices in each row has similar feature activations at the first hidden layer and therefore all subsequent hidden layers. To confirm this empirically, we randomly generated 2000 examples from each discovered polytope (by taking random convex combinations of the five polytope vertices) and checked the predicted class label and confidence of the prediction. As shown in Table 2, 100% of the 2000 examples are classified as the same class as the target example with extremely high confidence (1.0).

Interestingly, for the polytope in the first row, even though all polytope vertices are clearly 7s, the neural net should not classify examples with the polytope with the same confidence as it does on the target image, since the images in the polytope are clearly harder to discern. In this case, while the classification decision is correct, the confidence is not. For the polytope in the second row, the polytope vertices are clearly not 7s, yet the neural net still classified them as 7s with very high confidence. In this case, both the classification decision and the confidence are incorrect. Furthermore, all intermediate activations of the polytope and the target example are similar, which means that the neural net sees almost no difference at all between the target example and examples within the polytope.

We now turn our attention to the size of the polytopes. We report the average distance in Table 2, and compare it to the average distance between pairs of arbitrary images from the dataset. It turns out the average distance between polytope vertices is 12-13% of the average distance between images in the dataset, indicating that the polytope is quite large, even though it may not appear so visually, because humans are not adept at detecting differences at high frequencies.

| Target | Polytope vertices |
|---|---|
| 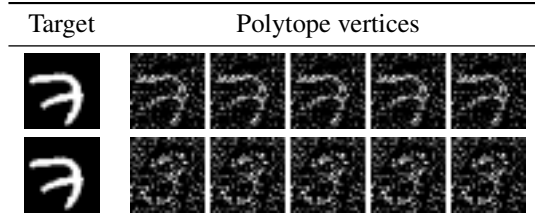 | |

Table 1: Two polytopes that collide with the same target example (an image of the digit 7). To the neural net, no example within either polytope is distinguishable from the target example.

We also applied thew proposed method to the LeNet-5 (LeCun et al.) convolutional network; the details and results are included in the appendix.

| Row# | % of successful collisions | Top class probability | | Average distance between images | |
|---|---|---|---|---|---|
| | | Target | Interpolated samples | MNIST | Optimized samples |
| 1 | 100% | 1.0 | 1.0 | 10.21 | 1.33 |
| 2 | 100% | 1.0 | 1.0 | | 1.28 |

Table 2: Quantitative results of MNIST experiments. First column corresponds to the row number of the experiments in Table 1. Second column shows the probability of randomly sampled images in the polytope being classified in the same class as the target image. Third column shows both the top class confidence for both the target and the samples. Fourth column compares the average $l_2$ distance between images from MNIST dataset with the average distance between images from the 5 optimized polytope vertices.

## 3.2 ImageNet Dataset

We now perform the same experiment on ImageNet. We use a pre-trained VGG-16 net (Simonyan & Zisserman (2014)) that achieves a 92.7% top-5 test accuracy. In our experiments, we tried to find feature collisions at the *fc6* layer.

We visualize the results in Table 3 and present the quantitative results in Table 11. For reasons of space, additional results are found in the appendix. As shown in Table 11, the neural net classifies all 2000 randomly chosen examples from each polytope into the category of the target example, again with very high confidence ($0.963 - 0.996$). In the case of ImageNet, the polytopes are even larger relative to the average distance between images from the dataset: the average distance between polytope vertices is now $12 - 24\%$ of the average pairwise distance between images, though the differences between the polytope vertices are not visually apparent for the same reason as above.

| Target | Polytope vertices |
|---|---|
| 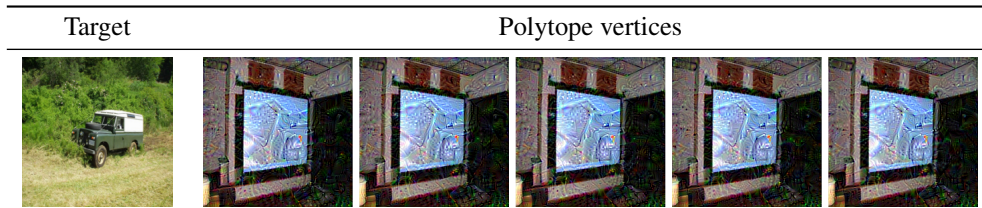 | |

Table 3: Colliding polytope on ImageNet that is found when initializing from another image. All examples within the polytopes collide with the target image.

| Row# | % of successful collisions | Top class probability | | Average distance between images | |
|------|---------------------------|-----------------------|---|-------------------------------|---|
| | | Target | Interpolated samples | ImageNet | Polytope vertices |
| 1 | 100% | 0.993 | 0.995 | 37538.22 | 4906.56 |

Table 4: Quantitative results corresponding to the results shown in Table 3.

## 4 More Perceptually Different Polytope Vertices

Do the colliding examples found in the previous section matter? One could argue that there is nothing to worry about, because if the different colliding examples are not perceptually different to a human, perhaps it is fine if a neural net cannot tell the difference between the different colliding examples. [5]

In this section, we demonstrate that colliding examples can still be found even when we bias the search towards perceptually different colliding examples. To this end, we constrain the search space to compositions of existing image patches, which are more perceptually different because changes in the way different patches are composed usually include changes in coarser details, which are more visually apparent.

More concretely, we parameterize the space of images in terms of compositions of existing image patches rather than raw pixel values, where the parameters specify how to compose different image patches. Because the space of image patches is much larger than the space of pixels, we have to constrain our set of patches to only those that could potentially be useful for solving our optimization problem.

We construct such a set by picking an image to start from, and then extract equal-sized patches from that image at different spatial locations. For each patch, we then retrieve the $k$ most similar patches from the dataset consisting of all patches extracted from images in the dataset. We then choose the solution space to be the space of possible convex combinations of these patches and optimize over the coefficients for combining the patches. More details are included in the appendix.

### 4.1 Results

We perform the experiment described above on the ImageNet dataset. We construct a dataset of patches by randomly selecting 10% of the images from each class and extracting $32 \times 32$ patches using a sliding window with constant pixel strides along horizontal and vertical directions. This yielded a dataset containing $37,011,074$ patches. To retrieve the $k$-nearest neighbours from this dataset, we use Prioritized DCI (Li & Malik, 2017), which can find the nearest neighbours in this dataset in five minutes.

We parameterize the coefficients for combining patches using a softmax over different possible patches, so that the coefficients at every pixel location lie within the range of $(0, 1)$ and sum up to 1.

Table 5 shows the results of our experiments. For reasons of space, additional results are found in the appendix. As shown, the differences between the polytope vertices are more obvious (for a guide on where to direct attention to, take a look at the rightmost column, which shows the locations in where the differences are most apparent). Yet, as shown in Table 6, 100% of the examples inside this polytope are classified the same as the target image with high confidence. This demonstrates that it is possible to find an *arbitrarily many* perceptually different examples (obtained by sampling from different locations in the polytope) that all look nearly identical to a neural net.

Table 6 also shows the average pairwise distance between the polytope vertices. It is interesting to note that the distance between the polytope vertices is similar in magnitude to the distance between the polytope vertices found previously in Table 11. This suggests that colliding polytopes where different points are perceptually different are not necessarily *larger* in volume than colliding polytopes where the differences are not as obvious. This suggests that the colliding polytopes found using the vanilla method are not necessarily insignificant – it's just that they seem less significant to the human eye.

| Target | Polytope vertices | | | | | Indicator |
|--------|-------------------|---|---|---|---|-----------|

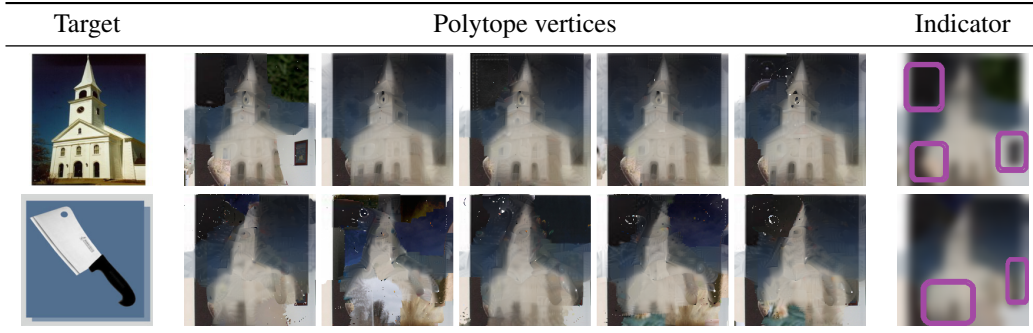

Table 5: ImageNet results, where the search space is constrained to compositions of image patches, thereby leading to more perceptually different polytope vertices. First row shows vertices optimized from an image of a church that share feature collisions with itself. Second row shows vertices optimized from the same image of a church but share feature collision with a cleaver. Images in the rightmost column highlights the differences between the vertices of the polytope using purple boxes.

| Row# | % of successful collisions | Top class probability | | Average distance between images | |
|------|-----------|--------|---------------------|----------|-------------------|
| | | Target | Interpolated samples | ImageNet | Polytope vertices |
| 1 | 100% | 0.952 | 0.877 | 37538.22 | 8597.34 |
| 2 | 100% | 0.999 | 0.800 | | 9587.92 |

Table 6: Quantitative results corresponding to the results shown in Table 5.

## 5   Discussion

While this paper focuses on the ReLU activation function, the observations are broadly applicable. First, the proposed method can be used to find colliding examples as long as there is some intermediate layer with ReLU activation, since once the features at that layer collides, features at later layers must collide as well.

Second, the observations hold approximately for any activation function that saturates, i.e. sigmoid, tanh or ELU, since in the saturating region, the pre-activation can be changed substantially without significantly changing the post-activation.

The phenomenon of feature collisions can be used in various applications. Some possible applications that we can foresee are below:

1. **Representative Data Collection:** The size of the colliding polytope around a training example can be used to discover regions of the data space where insufficient training examples have been collected. More concretely, if the size of a colliding polytope around a training example is large, then the neural net could be over-generalizing in the neighbourhood of that example, and so the model may not be accurate in this neighbourhood. This can be used to inform the end-user whether the prediction in this neighbourhood should be trusted, or to guide data collection, so that more examples in this neighbourhood are collected in the future.

2. **Design of Regularizers:** The insight our method reveals can lead to the design of regularizers that mitigates undesirable over-generalization. For example, one could try to minimize the size of the colliding polytopes, by discouraging the hyperplanes associated with each hidden unit from being near-collinear (i.e. having highly positive cosine similarity) with other hyperplanes. This can be also used to guide architecture selection.

3. **Identification of Vulnerable Training Examples:** The proposed method can identify the training examples that a neural net depends most on, which could have large colliding polytopes around them. This can help detect outliers and training examples that could have been mislabelled or adversarially tampered with, or legitimate training examples that could be vulnerable to manipulation due to how much the neural net depends on them.

# 6 Related Work

There have been several lines of work that use iterative optimization to find noteworthy input examples. One line of work is on adversarial examples, where the goal is to find a small perturbation to a source input example so that it is misclassified. Tatu et al. (2011) proposed using projected gradient descent to find a perturbed version of an example with similar SIFT features as an example from a different class. Szegedy et al. (2013) demonstrated a similar phenomenon in neural nets, where an adversarially perturbed example can be made to be classified by neural nets as an example from any arbitrarily class. Nguyen et al. (2015) further shows that it is easy to generate images that do not resemble any class, but are classified as a recognizable object with high confidence. Kurakin et al. (2016) demonstrated that even when adversarial examples are printed on a sheet of paper, they are still effective at fooling neural nets. Athalye & Sutskever (2017) even managed to 3D print adversarial example models. More recently, Shafahi et al. (2018) showed that it is possible to adversarially perturb examples so that their features are close to the features of a completely different example.

Similar techniques have also been used for a different purpose, namely to understand what input image would cause the activation of a particular neuron in neural nets to be high, which is known as *activation maximization* (Erhan et al., 2009). This technique can be applied to either a neuron in the output layer (Simonyan et al., 2013) or a hidden layer (Erhan et al., 2009; Yosinski et al., 2015). A related line of work, known as *code inversion* (Mahendran & Vedaldi, 2015; Dosovitskiy & Brox, 2016), aims to find an input image whose entire activation vector after a particular layer is similar to the activation vector of a particular real image. Unlike the adversarial example literature, the goal of this body of work is to find an *interpretable/visually recognizable* image that allows for the visualization of the kinds of images that would either cause a particular neuron to activate or all neurons in the same layer to exhibit a particular pattern. Therefore, a regularizer is usually included in the objective function that favours images that are more natural, e.g.: those that are smooth and do not have high frequencies. Surprisingly, the images that are found often bear resemblance to instances in the target class. It has been conjectured (Mahendran & Vedaldi, 2015; Nguyen et al., 2016) that the inclusion of this regularizer would explain the apparent discrepancy between the findings of the adversarial example literature and the code inversion literature – code inversion is simply not finding adversarial examples because having high-frequency perturbations is penalized by the regularizer. Our experiments show that this may in fact not be true, since we are able to successfully find examples that do not have high frequencies but clearly do not resemble any instance of the target class. We conjecture this may be because points near the centre of the polytope may not be reachable using a naïve loss function that only penalizes the difference between post-activations; this is because the gradient becomes zero as soon as the input example is moved into the polytope. As a result, the solution found using gradient descent will usually be near the boundary of the polytope, which may happen to resemble objects in the target class.

Concurrent work (Jacobsen et al., 2018) also explores a similar theme. One difference with this work is that they only consider collisions at the level of the classification predictions, rather than collisions at the level of lower-level features.

# 7 Conclusion

In this paper, we have shown theoretically that polytopes of examples sharing the same feature embeddings could exist due to the properties of ReLU activation functions. We developed a method for finding such polytopes and demonstrated empirically that they do in fact exist in commonly used neural nets. Somewhat surprisingly, in Section 4.1, we demonstrated that even after constraining examples to be compositions of image patches, these polytopes still exist. Furthermore, the vertices of the polytope appear perceptually different, which shows that interpolations in this polytope can all be misclassified even though each interpolation is visually distinctive.

### Acknowledgements

This work was supported by ONR MURI N00014-14-1-0671. Ke Li thanks the Natural Sciences and Engineering Research Council of Canada (NSERC) for fellowship support.

## Footnotes

[3]If the polytope is unbounded, we can still generate new colliding examples this way, but there could be colliding examples that are not convex combinations of the vertices.

[4]https://github.com/aymericdamien/TensorFlow-Examples

[5]Note that this argument cannot be made for the difference between the target example and a colliding example; the target example is clearly different from a colliding example to a human, but triggers nearly identical activations as the colliding example. This is clearly undesirable.

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
