[Supplementary Material]

# Appendix A    Derivation of Condition in Proposition 1

Let $\mathbf{b}^+$ and $\tilde{\mathbf{y}}^{t+}$ denote the subvectors of $\mathbf{b}$ and $\tilde{\mathbf{y}}^t$ respectively containing only the components where the corresponding components in $\tilde{\mathbf{y}}^t$ are positive. Consider a point $\mathbf{x}$ in the subspace defined by the conjunction of equality constraints. By definition, this point must satisfy $\mathbf{W}^+\mathbf{x} + \mathbf{b}^+ = \tilde{\mathbf{y}}^{t+}$. Now consider each inequality constraint, which is a half-space. If the dividing hyperplane of the half-space is *not* parallel to the subspace, then the projection of the half-space onto the subspace is a half-space in the subspace. If it is parallel, there are two possibilities: either the subspace lies completely inside the half-space, or outside of it. If it is inside, the projection of the half-space onto the subspace is just the subspace. If it is outside, the projection is the empty set. However, this is not possible because the target example must be in the projection. Therefore, the subspace must lie completely in the half-space.

So, checking whether the projection of the half-space onto the subspace is the entire subspace amounts to checking whether the dividing hyperplane is parallel to the subspace.

Since translations preserve parallelism, we can arbitrarily translate the subspace. For simplicity, we can translate the subspace so that it always interesects with the origin. This subspace is characterized by $\mathbf{W}^+\mathbf{x} = \mathbf{0}$. Then if it is parallel to the dividing hyperplane of the half-space, every point must lie on the dividing hyperplane, since the hyperplane also intersects with the origin.

When this happens for all the inequality constraints, then $\forall \mathbf{x}$ such that $\mathbf{W}^+\mathbf{x} = \mathbf{0}$, $\mathbf{W}^-\mathbf{x} = \mathbf{0}$. In other words, $\ker(\mathbf{W}^+) \subseteq \ker(\mathbf{W}^-)$.

# Appendix B    Implementation Details for Patch-Based Experiment

We first extract patches using a sliding window, of which the size is the patch size $n \times n$. Assume that we have $m$ patches, then by searching for nearest neighbours of these patches in the dataset, $m$ nearest neighbours will be found. For each nearest neighbour, as shown in Figure 3a, we place it along with the surrounding pixels on a blank image at the same position from which the query patch was extracted.

Let $s$ denote the width of the band of pixels surround the patch and let $\hat{n} = n + 2s$ be the total size of the patch and the surrounding pixels. A schematic diagram of various regions of interest in the source image is shown in Figure 3b. Now let $Q = \{q^0, q^1, \cdots, q^{m-1}\}$ denote the source images and $C = \{(c_x^0, c_y^0), \cdots, (c_x^{m-1}, c_y^{m-1})\}$ denote the centre coordinates of the patches in source images. We combine the source images together with linear interpolation to generate the composite image $I$, where

$$I_{i,j} = \frac{\sum_{k \in A_{i,j}} (\hat{n}/2 - |i - c_x^k|)(\hat{n}/2 - |j - c_y^k|)q_{i,j}^k}{\sum_{k \in A_{i,j}} (\hat{n}/2 - |i - c_x^k|)(\hat{n}/2 - |j - c_y^k|)}$$

Here $A_{i,j}$ denotes the set of all source images that overlap with position $(i, j)$ in the blank image when positioned for pasting, that is

$$A_{i,j} = \{k | \hat{n}/2 - |i - c_x^k| \geq 0, \hat{n}/2 - |j - c_y^k| \geq 0\}$$

We can further write

$$I_{i,j} = \sum_{k \in A_{i,j}} p_{i,j}^k q_{i,j}^k$$

where

$$p_{i,j}^k = \frac{(\hat{n}/2 - |i - c_x^k|)(\hat{n}/2 - |j - c_y^k|)}{\sum_{k \in A_{i,j}} (\hat{n}/2 - |i - c_x^k|)(\hat{n}/2 - |j - c_y^k|)} \tag{1}$$

is called the control parameter for the $k$th source image at position $(i, j)$. Therefore the composite image can be now defined as

$$I = G(P, Q)$$

where $P$ denotes the set of control parameters and $Q$ denotes the set of source images.

Now instead of optimizing w.r.t. the pixels, we optimize w.r.t. the control parameters. This entire process is illustrated in Figure 2.

Figure 2: A diagram that details the procedure for running the patch-based experiment.

(a) Query patch       (b) Nearest Neighbor       (c) Source image

(a) A diagram showing how the retrieved patches are obtained and used. (a) is the query image and the green rectangle labels a query patch. We obtain the $k$-nearest neighbours (in the conv1_2 feature space of a pretrained VGG-16 net), one of which is shown by inside the green rectangle in (b). (c) is the so-called source image, which is a blank image where the retrieved patch is placed at the same spatial position where the query patch was extracted. The pixels within a fixed distance from the border of the retrieved patch are also padded around the the patch in the source image.

(b) Sizes of various regions that comprise the source image.

To encourage the patches used in the composite image to be contiguous, we use a regularizer that penalizes the difference between weights on adjacent pixels of the same source image. More precisely, if there are $K$ source images of size $M \times N$, the objective function would be

$$\mathcal{L}^i(P) = \alpha\mathcal{L}_{pos}(G(P,Q)) + \beta\mathcal{L}^i_{neg}(G(P,Q)) + \gamma\mathcal{R}(P) \tag{2}$$

where $\mathcal{R}(P)$ is the contiguity regularizer, which is defined as

$$\mathcal{R}(P) = \sum_{k=1}^{K}\left(\sum_{i=1}^{M}\sum_{j=1}^{N-1}|p^k_{i,j} - p^k_{i,j+1}| + \sum_{j=1}^{N}\sum_{i=1}^{M-1}|p^k_{i,j} - p^k_{i+1,j}|\right)$$

## Appendix C    Comparison of Polytope and Ball

To demonstrate that the polytope we find cannot be trivially replaced with a ball of similar size, we consider a ball centred at the centroid of the polytope whose radius is the minimum distance from the centroid to a corner of the polytope. We then randomly sample points inside the ball and check

whether they are classified as/collide with the target class. The percentage of points that collide with the target class is shown in Table 7. Whereas the percentage of points inside the polytope that collide with the target class is 100%, the percentage of points inside the ball that collide with the target class is much smaller, as shown. This demonstrates that samples from a ball of similar size does not always collide with the target class, unlike in the case of the polytope.

|  | % of successful collisions |
|---|---|
| 1st row in Table 5 | 9.2% |
| 2nd in Table 12 | 0.0% |
| 1st row in Table 5 | 24.8% |
| 2nd row in Table 12 | 3.5% |

Table 7: Percentage of points from a similarly sized ball centred at the centroid of the polytope that are classified as the target class.

## Appendix D    Additional Experimental Results

We further conducted experiments on MNIST using the LeNet-5 convolutional network, which consists of three convolutional layers. The trained model achieves a test accuracy of 98.0%. We find feature collisions at the first fully connected layer. The results are shown in Table 8 and Table 9.

| Target | Polytope Corners | | | | |
|---|---|---|---|---|---|

Table 8: Two polytopes that collide with the same target example (an image of the digit 7).

| Row# | % of successful collisions | Top class probability | | Average distance between images | |
|---|---|---|---|---|---|
| | | Target | Interpolated samples | MNIST | Optimized samples |
| 1 | 100% | 1.0 | 1.0 | 10.21 | 1.24 |
| 2 | 100% | 1.0 | 1.0 | | 3.08 |

Table 9: Quantitative results corresponding to the results shown in Table 8.

Table 10: Colliding polytope on ImageNet when initializing from the target image. All examples within the polytopes collide with the target image.

| Row# | % of successful collisions | Top class probability | | Average distance between images | |
|---|---|---|---|---|---|
| | | Target | Interpolated samples | ImageNet | Polytope corners |
| 1 | 100% | 0.963 | 0.982 | 37538.22 | 8166.43 |

Table 11: Quantitative results corresponding to the results shown in Table 10.

Table 12: Results on ImageNet when search space is constrained to compositions of image patches. First row shows corners optimized from an image of a trimaran that share feature collision with an obelisk. Second row shows corners optimized from an image of an eggnog that share feature collision with an image that is misclassified as an "eraser". Images in the rightmost column highlights the differences between the corners of the polytope using purple boxes.

| Row# | % of successful collisions | Top class probability | | Average distance between images | |
|---|---|---|---|---|---|
| | | Target | Interpolated samples | ImageNet | Polytope corners |
| 1 | 100% | 0.997 | 0.992 | 37538.22 | 5909.37 |
| 2 | 100% | 0.579 | 0.585 | | 6074.86 |

Table 13: Quantitative results corresponding to the results shown in Table 12.

# Appendix E  Intermediate Images During Optimization

To better understand the optimization process of finding the polytope we visualize the intermediate images during optimization. Figures 4, 5 and 6 each show the intermediate images at different points of the optimization process for two corners randomly chosen from the five that are shown in Figures (b), (c) and (d) in Table 5 respectively.

(a)

(b)

Figure 4: Intermediate images for Figure (b) in Table 5

(a)

(b)

Figure 5: Intermediate images for Figure (c) in Table 5

(a)

(b)

Figure 6: Intermediate images for Figure (d) in Table 5