[Reviews · NeurIPS 2019]

Reviewer 1



This work is outside by immediate research area, but I found it interesting. I saw examples along the lines of those shown in Figure 3 at least fifteen years ago illustrating the finite capacity of ANNs and the importance of architecture. In the particular context considered here, this is shown very explicitly in the context of the input layer when using ReLU activations. The thresholding nature of these activation functions indeed allow a precise characterization of the behavior. I find it difficult to believe that experts in the field are not aware of this potential behavior but the paper develops the idea considerably and seems interesting. It seemed to me that one possible use of the developed algorithm would be to provide a post hoc assessment of the adequacy of a trained neural network: given the trained network is there a collision polytope of sufficiently diverse inputs that one would be unhappy to depend upon the network? I was surprised not to see this in the discussion. The example in Section 3.1 seems to be somewhat simpler than the state of the art in this area: does one find similar behavior with more complex conv nets with appropriate input features? l216 founding -> found? It might be interesting to comment on the extent to which the findings might transfer approximately to other activation functions -- sigmoidal and tanh activations, for example, map large parts of the state space to values very close to +/-1. Details: l83: the intersection of halfspaces is in general a convex polytope, but can easily be empty. I'd have liked to see some discussion of if/when that situation might arise. l87 infinitely colliding examples -> infinitely many colliding examples? I thank the reviewers for their response, particularly developing some possible applications on the work, which seem to me to strengthen the work somewhat, and for providing numbers for a more sophisticated network.

Reviewer 2



I believe the phenomenon identified here seems interesting and worth considering. The authors point out the work of Jacobsen et al. which addresses a similar topic, but as concurrent work this does not diminish the originality of the paper here. (And moreover Jacobsen et al. address network invariance from the perspective of the classification boundary and not necessarily with respect to the internal features.) The paper is easy to follow in most parts. The introduction in particular very clearly defines and unpacks the phenomenon of interest. The Method section would benefit from slightly more care in its technical details. For instance, possible edge-cases occur to me in the claim that "the intersection of all the sets listed above is a convex polytope in at least a d-n_p dimensional subspace". I believe this would not hold for instance in the case that several (w_i, b_i) define parallel but distinct hyperplanes? Likewise, the implications of the objective function (3) are not clear to me. Is the true minimizer of this always necessarily a corner of the polytope, or do we simply hope it is close to one? From the comment on line 159 that the "intermediate activations of the polytope and the target are similar", i.e. not *exact*, I suspect the latter, but it isn't obvious. In any case, the clarity of this section could benefit from some small Propositions in the appendix that address these and similar points. I was also not completely clear on the methodology used for the experiment in Section 4.1. Is the same database of patches that is extracted from the original dataset used to define the possible patches at each location in the new image? E.g. would the softmax described in lines 203-204 have 37011074 outputs? Or do you only consider patches drawn from the same locations in the original images? In summary, the paper explores an interesting phenomenon that has potential practical implications for neural network robustness. However, it could benefit from some points of greater clarity, particularly in sections 2 and 4. === UPDATE ============ Having read the authors' response and the other reviews, my opinion is largely unchanged. I believe the topic is potentially of some interest to the community, although I agree with the other reviewers that it is a bit incremental. I also still think it would benefit from greater clarity - the authors have at least promised to address this, although without seeing a revised version I can't confirm that this is not still an issue. Overall, my score is still at a 6 (weak accept). I would give the paper a 5.5 if I could.

Reviewer 3



This paper explores the problem of feature collision in ReLU networks --- where examples contained within large polytopes in the input space all map to the same (or approximately the same) feature space. The paper shows that this is driven by the characteristics of the ReLU function, namely that they are many-to-one over negative activations. The existence of such large polytopes is an interesting and complimentary result to the recent work on adversarial examples (i.e. excessive invariance versus excessive sensitivity). In terms of methodology, the paper gives a well reasoned and executed solution for finding the polytopes. The proposed method enables one to quickly find polytopes that all map to approximately the same feature space for (what empirically appears to be practically any) arbitrary input examples. Overall, the paper is very well-written and the approach is interesting and novel. Though the analysis of this behavior and the findings regarding the ubiquity of these invariant polytopes is stimulating in its own right, my only concern is that I am still slightly unsure of the significance of these results. This doesn’t seem to be a goal of the paper, but I didn't find the discovered polytopes to be that convincingly distinctive. They still seemed visually similar (albeit at low human frequencies) or have clear artifacts (even in the image-patch-constrained experiments), so adversarial generation doesn’t quite seem to be a clear next step. Additionally, though a qualitative observation, it’s not completely obvious that the polytopes given in the figures should even be considered distinctly that bad for mapping to the same class/~features. Clearly there are artifacts of the target image in the polytope corners (even in Table 5). However, I do agree to some extent that in general it can be quite undesirable behavior for them to have the same confidence scores. Finally, I wonder if some additional analysis into the polytopes could be useful for interpretability or deriving rationales (particularly in the case of natural language, where using words instead of image patches might lead to very distinguishable results). Minor comments: line 32: in fact, there could be a line 91: I would move the comment in parentheses to a footnote line 174: "the polytope that even larger relative" needs fixing line 216: the colliding polytopes found === After Response === Thank you for your response and clarification. The suggested possible ramifications of these findings (data representation, regularization) indeed seem reasonable (if a bit optimistic).

[Author Response · NeurIPS 2019]

We thank the reviewers for their feedback. In our paper, we show the existence of different examples that map to the same feature activation in a neural net with ReLU activations, which can be viewed as, in some sense, the *opposite* of adversarial examples, and demonstrate that there could be infinitely many such examples lying in convex polytopes. All reviewers found the paper "interesting", and various reviewers commented that "the phenomenon identified here seems interesting and worth considering", "the paper gives a well reasoned and executed solution" and "the approach is interesting and novel".

R1 and R3 asked about the possible applications of the findings, and we briefly discuss several different ways in which our observation can be used below (we will discuss these more extensively in the camera-ready):

1. **Representative Data Collection:** The size of the colliding polytope around a training example can be used to discover regions of the data space where insufficient training examples have been collected. More concretely, if the size of a colliding polytope around a training example is large, then the neural net could be over-generalizing in the neighbourhood of that example, and so the model may not be accurate in this neighbourhood. This can be used to inform the end-user whether the prediction in this neighbourhood should be trusted (as suggested by R1), or to guide data collection, so that more examples in this neighbourhood are collected in the future.

2. **Design of Regularizers:** The insight our method reveals can lead to the design of regularizers that mitigates undesirable over-generalization. For example, one could try to minimize the size of the colliding polytopes, by discouraging the hyperplanes associated with each hidden unit from being near-collinear (i.e. having highly positive cosine similarity) with other hyperplanes. This can be also used to guide architecture selection.

3. **Identification of Vulnerable Training Examples:** The proposed method can identify the training examples that a neural net depend most on, which could have large colliding polytopes around them. This can help detect outliers and training examples that could have been mislabelled or adversarially tampered with, or legitimate training examples that could be vulnerable to manipulation due to how much the neural net depends on them.

We respond to the main remarks by each individual reviewer below:

**R1:**　We agree that the existence of feature collisions should be well-known (and it certainly was to us as authors), but found from conversations with others that it is not actually known to many people. So, the purpose of this paper is to highlight this phenomenon, which we hope will spark further investigation in this direction. It is possible that this phenomenon was pointed out in a previous paper; we weren't able to find an example of this in our literature search, but would be happy to include a reference to it if you are aware of such a paper.

For the results in Sect. 3.1, yes, we do find similar behaviour with more complex convnets. We tried the same experiment on LeNet, and found that 100% of the sampled input within the polytope are successful collisions and that they are all classified the same as the target example with a confidence of 100%. The average distance between the polytope corners is actually larger – it is 1.24 when the source and target examples are the same and 7.45 when the source and target examples are different (which is quite large, since the average distance between MNIST examples is 10.21).

Thanks for the suggestion – the findings do hold approximately for any activation function that saturates, i.e. sigmoid, tanh or ELU. We will comment on this in the camera-ready. If the polytope is around a target example, then it would not be empty by construction because the target example must be contained in the polytope. However, it is possible that when the activations for all hidden units are positive, the polytope only contains a single point, namely the target example.

**R2:**　Thank you for your suggestion! We will formalize the observations in Sect. 2 as propositions in the camera-ready.

To respond to your questions, if all hyperplanes are parallel, the polytope would still be convex, but would not be bounded. As mentioned on L84 and L91-93, the implication is that there could be colliding examples that we would not be able to generate, but we can still generate colliding examples from a (bounded) subset.

If we were to enforce the equality constraints on the hidden units with positive pre-activations and the hidden units with non-positive pre-activations other than the current hidden unit $i$ strictly, and assuming the hyperplane associated with hidden unit $i$ is not parallel to the hyperplanes of other hidden units, then the solution is an exact corner of the polytope. The reason why the activations are not identical to the target example is because the equality constraints are not enforced strictly.

For Sect. 4.1, the softmax is over just over the $k$-nearest neighbours of the patches of the source image at each spatial location, where $k$-NN is performed over the dataset of 37,011,074 patches.

**R3:**　Thanks for the suggestion! We agree feature collisions on language data could be easily distinguishable perceptually and would be interesting to explore.

[Meta-Review · NeurIPS 2019]

All reviewers are positive about the paper. The authors present an interesting geometrical analysis of deep-net feature representations. The paper introduces a notion of collision polytope and provides algorithms to find vertices of collision polytopes. Interesting illustrations of the proposed notions and algorithms are presented. We recommend to take the reviewers' comments and suggestions into account while preparing the camera ready final version of the paper. Accept.